# Mesenchymal Stem Cell-Derived Extracellular Vesicles: A Potential Therapy for Diabetes Mellitus and Diabetic Complications

**DOI:** 10.3390/pharmaceutics14102208

**Published:** 2022-10-17

**Authors:** Fengtian Sun, Yuntong Sun, Feng Wu, Wenrong Xu, Hui Qian

**Affiliations:** Jiangsu Key Laboratory of Medical Science and Laboratory Medicine, Department of Laboratory Medicine, School of Medicine, Jiangsu University, Zhenjiang 212013, China

**Keywords:** extracellular vesicles, mesenchymal stem cell, diabetic mellitus, diabetic complications, engineering

## Abstract

As a novel cell-free strategy, mesenchymal stem cell-derived extracellular vesicles (MSC-EVs) inherit the therapeutic potential of donor cells, and are widely used for the treatment of many diseases. Increasing studies have shown that MSC-EVs transfer various bioactive molecules to create a beneficial microenvironment, thus exerting protective roles in diabetic mellitus (DM) and diabetic complications. To overcome the limitations of natural MSC-EVs such as heterogeneity and insufficient function, several modification methods have been established for constructing engineered MSC-EVs with elevated repairing efficiency. In this review, the PubMed library was searched from inception to August 2022, using a combination of Medical Subject Headings (MeSH) and keywords related to MSC-EVs, DM, and diabetic complications. We provide an overview of the major characteristics of MSC-EVs and summarize the recent advances of MSC-EV-based therapy for hyperglycemia-induced tissue damage with an emphasis on MSC-EV-mediated delivery of functional components. Moreover, the potential applications of engineered MSC-EVs in DM-related diseases therapy are discussed by presenting examples, and the opportunities and challenges for the clinical translation of MSC-EVs, especially engineered MSC-EVs, are evaluated.

## 1. Introduction

Diabetes mellitus (DM) is a group of metabolic diseases characterized by chronic hyperglycemia as a result of defects in insulin secretion and/or function [1]. Accompanied by population aging and dietary change, the global prevalence of DM rises steadily [2]. Type 1 DM usually occurs in adolescents and is featured by the absolute deficiency of insulin secretion due to pancreatic islet β-cells reduction [3]. Type 2 DM is common in elder and obese patients because of the decreased insulin sensitivity of peripheral tissues and reduced insulin secretion of pancreatic islet β-cells [4]. Persistent hyperglycemia is prone to cause many diabetic complications such as diabetic nephropathy, diabetic retinopathy, diabetic neuropathy, and diabetic ulcer [5]. Currently, the utilization of therapeutic insulin and hypoglycemic agents provides possibilities to effectively control the blood glucose levels of diabetic patients, whereas the long-term application of these drugs may induce severe side effects and cannot prevent the progress of diabetic complications [6,7]. Therefore, there is an urgent need to develop novel strategies to treat DM and its related chronic complications.

Mesenchymal stem cell (MSC) is recognized as an important source cell in regenerative medicine due to its capacity for self-renewal and multidirectional differentiation [8]. MSC can be isolated from various tissues including bone marrow, adipose tissue, umbilical cord, gingiva, and synovium [9]. Increasingly, studies have shown that MSC transplantation displays therapeutic value to ameliorate many refractory diseases by replacing injured cells and secreting growth factors and anti-inflammatory cytokines [10,11]. Importantly, the discovery that MSC establishes a beneficial microenvironment to accelerate the regeneration of pancreatic β-cells promotes the idea of using them as new therapeutics for DM [12,13]. However, MSC is reported to possess the same stemness-related gene phenotypes as mesenchymal tumor cells, implying that MSC may determine early tumor formation [14]. Moreover, MSC administration can promote the chemo-resistance and invasion of tumor cells [15,16], thus limiting its wide application.

Recent evidence has revealed that the therapeutic effects of MSC are mediated by the paracrine pathway, mainly through the release of extracellular vesicles (EVs) [17]. “EVs” is a generic term for various subtypes of membrane vesicles secreted by cells. EVs contain various bioactive molecules including proteins, nucleic acids, and lipids, and transport them from donor cells to recipient cells [18]. Accumulating studies have suggested that EVs represent a new manner of intercellular communication and contribute to the regulation of many pathophysiological processes [19,20]. In addition, the lipid bilayer membrane structure endows EVs with great stability in circulation and protects their cargo from degradation [21]. Recently, the cell-free strategy based on MSC-derived EVs (MSC-EVs) for tissue regeneration has attracted considerable attention. MSC-EVs not only inherit the ability of MSC to repair damaged tissues, but also avoid cell therapy-induced limitations such as cell senescence, low cell survival, and tumorigenicity [22,23]. Moreover, the engineering of MSC-EVs can further enhance their therapeutic efficiency, bioactivity, and yield through various modification methods. MSC-EV-mediated protective functions in DM and its chronic complications have been evaluated in many preclinical studies and achieved encouraging results.

In this review, we summarize the recent advances of natural or engineered MSC-EVs in the treatment of DM and several major diabetic complications and focus on the role of MSC-EV-delivered bioactive molecules in mediating tissue protection. We also discuss the modification strategies of MSC-EVs and evaluate the future opportunities and challenges of this field.

## 2. Biogenesis, Contents, and Characteristics of MSC-EVs

EVs are a heterogeneous group of membrane-structured vesicles secreted by living cells. Since the specific markers of EV subtypes warrant further discussion and investigation, the International Society for Extracellular Vesicles (ISEV) recommends researchers use operational terms for defining EVs according to physical characteristics (size and density), biochemical composition, or descriptions of conditions or cell of origin [24]. For instance, EVs can be divided into small EVs (<100 nm or <200 nm) and medium/large EVs (>200 nm). Previous studies have usually classified EVs into exosomes, microvesicles, and apoptotic bodies [25]. However, strictly speaking, they should only be used when data revealing the formation processes of EVs are provided. Exosomes, the smallest subpopulation of EVs with a size ranging from 30–200 nm, are generated through a series of complex processes including endocytosis, endosome maturation, and multivesicular bodies (MVBs) formation [26]. After the fusion of MVBs with plasma membranes, exosomes are released into extracellular space. Microvesicles, with a diameter of 100–1000 nm, are derived from the direct outward budding of plasma membranes [27]. Apoptotic bodies (greater than 1000 nm in diameter) are formed by cells after apoptosis [28]. In this review, the umbrella term “EVs” is used to refer to these vesicles on the basis of Minimal Information for Studies of Extracellular Vesicles 2018 (MISEV2018) guidelines. Initially, EVs were considered as a way for the excretion of cell waste. With the rapid development of high-throughput sequencing technologies, recent studies have revealed that EVs carry multiple bioactive molecules including proteins, nucleic acids, and lipids [29]. Several proteins are considered the markers of EVs such as the tetraspanin protein family (CD9, CD63, and CD81), proteins involved in membrane fusion (Rab5, Rab7, Annexin A1, Annexin A2, and Annexin A7), MVBs synthetic proteins (Alix and TSG101), chaperone proteins (HSP60, HSP70, and HSP90), and phospholipases [30]. Moreover, EVs derived from different donor cells contain many specific proteins, which may determine their biological functions. For instance, Zhang et al. reported that MSC-EVs but not human lung fibroblasts-derived EVs promote cutaneous wound healing as a result of the specific expression of Wnt4 in MSC-EVs [31]. In addition, MSC-EVs also express MSC surface markers such as CD44, CD73, and CD90 [32]. Nucleic acids are another important component of EVs. Increasingly, studies have shown that EVs deliver mRNA, miRNA, lncRNA, circular RNA, genomic DNA, and mitochondrial DNA to modulate the biological behavior of recipient cells [33]. The lipids in EVs exhibit the ability to participate in various biological processes such as signal transduction, microenvironment regulation, and inflammation response [34,35]. Lipid-raft domains of the plasma membrane contribute to the early endosome formation through the endocytosis pathway, thus initiating the biogenesis process of exosomes [36]. Notably, the lipid bilayer membrane structure endows EVs with the potential to protect internal cargos in circulation [37]. EVs act as novel vehicles to mediate intercellular communication and transport bioactive molecules from their donor cells to distant recipient cells. The internalization of EVs involves several pathways including membrane fusion, receptor-ligand interaction, and endocytosis [38]. Among them, various endocytosis-related ways are considered the major mechanisms underlying the uptake of EVs by recipient cells, including clathrin-dependent endocytosis, caveolin-mediated uptake, macropinocytosis, phagocytosis, and lipid raft-mediated internalization [39].

Although the mechanisms responsible for the cargo sorting into EVs are still not fully understood, researchers have proposed several possibilities. Accumulating evidence has shown that the endosomal sorting complexes required for transport (ESCRT) pathway contributes to the loading of proteins into MVBs in a ubiquitin-dependent manner [40]. The ESCRT contains four distinct complexes named ESCRT-0, ESCRT-I, ESCRT-II, and ESCRT-III, together with the ATPase vacuolar protein sorting-associated protein 4 (VPS4). Each subcomplex has unique roles and exerts functions sequentially including cargo clustering (ESCRT-0), cargo binding (ESCRT-I, and ESCRT-II), vesicles maturation and constriction (ESCRT-III), and membrane scission (VPS4) [41]. In addition, it is reported that EVs can also be generated in the absence of ESCRT proteins, implying the existence of ESCRT-independent mechanisms [42]. Recent studies have suggested that ceramides and tetraspanin-enriched microdomains are also associated with cargo sorting into EVs [43]. The introduction of RNAs into EVs involves the help of a complicated RNA sorting system. RNA binding proteins (RBPs) display the ability to recognize specific RNAs with unique sequences and structures and then load them into EVs [44]. Moreover, adenylation and urylation at the 3′ ends of miRNAs, human antigen R, and argonaute 2 also promote RNA sorting into EVs [45,46]. However, the potential mechanisms by which DNAs can be packaged into EVs still need further investigation (Figure 1).

MSC-EVs can be isolated from the cell supernatants of MSC. According to the physical and chemical features of MSC-EVs, several isolation methods have been established such as ultracentrifugation [47], density gradient centrifugation [48], size exclusion chromatography [49], immunoaffinity capture [50], ultrafiltration [51], and polymer precipitation [52]. Each method has its own advantages and limitations (Table 1). Currently, there is still a lack of low-cost approaches to rapidly obtain MSC-EVs with high purity, good integrity, and high yield. Researchers need to utilize different extraction methods based on the experimental requirements.

The findings that MSC exerts therapeutic effects mainly through the paracrine pathway facilitate studies on the biological functions of MSC-EVs. Although MSC transplantation represents a feasible strategy for diabetic disease therapy, the therapeutic efficiency remains unsatisfactory. Intravascular infusion of MSC may result in vascular embolism because of its relatively large size [53]. MSC-based cell therapy contains the risks of immune rejection and tumor formation [54]. In addition, due to the rapid senescence of MSC, the large-scale production of MSC with high cell viability is costly [55]. By contrast, MSC-EV-mediated cell-free strategy has several advantages: (1) MSC-EVs exhibit the natural potential to cross physiological barriers such as blood-brain barrier and blood-retinal barrier as a result of their nanoscale size [56]. (2) MSC-EVs treatment reduces the risks of cell transplantation-induced immune rejection and tumorigenicity [57]. However, toxicological and safety evaluations still require further studies following the long-term administration of MSC-EVs. (3) MSC-EVs possess high biocompatibility and stability and can be rapidly absorbed by recipient cells to transfer bioactive molecules [58]. (4) MSC-EVs can maintain their cargo at low temperatures. Storage and transportation of MSC-EVs are more convenient [59]. (5) Proper modification can further improve the targeting property and repairing effects of MSC-EVs [60]. Therefore, these unique characteristics make MSC-EVs one of the promising candidates for regenerative medicine.

## 3. Natural MSC-EVs in the Treatment of DM and Diabetic Complications

### 3.1. Diabetes Mellitus

In the treatment of DM, MSC is the most common donor cell of EVs due to its intrinsic repairing value. Islet cell death is recognized as a key barrier to successful islet cell transplantation. MSC-EV-mediated delivery of protective molecules such as vascular endothelial growth factor (VEGF) and miR-21 is reported to protect isolated islets’ survival and enhance their viability [61,62]. Accumulating evidence has shown that the direct therapeutic role of MSC in DM is mainly mediated by the secretion of EVs. Favaro et al. revealed that MSC-EVs exhibit similar immunomodulatory functions to MSC in inducing the immature phenotype of dendritic cells from type 1 diabetic patients, thus inhibiting inflammatory T-cell response in islet tissues [63]. Moreover, the administration of MSC-EVs via the tail vein effectively prevents type 1 DM progress by suppressing the activation of antigen-presenting cells and the development of T helper cells [64]. In another study by Nojehdehi et al., MSC-EV treatment can enhance the number of islets and improve glycemic control by increasing the regulatory T-cell population in streptozotocin (STZ)-induced type 1 DM mice [65]. In addition to the immune regulation mechanisms, MSC-EVs also show the ability to directly affect glucose metabolism. Many studies focus on the potential of MSC-EVs in the improvement of β-cells due to their insulin-secreting function. In the STZ-induced type 2 DM rat model, Sun et al. found that MSC-EVs injection reverses hyperglycemia-induced glucose metabolism disorders and β-cells apoptosis, providing an alternative strategy for DM treatment [66]. Similarly, Cooper et al. demonstrated that MSC-EVs can increase islet number and β-cell mass to elevate the insulin level in the circulation, leading to decreased blood glucose levels in diabetic mice [67]. The results of Mahdipour et al. confirm the accumulation of MSC-EVs in the pancreas after intravenous injection, which promotes the regeneration of β-cells through the Pdx-1-dependent mechanism [68]. Furthermore, many attempts have been made to illustrate the therapeutic molecules in MSC-EVs. By miRNA sequencing, Sharma et al. identified that MSC-EVs transport multiple miRNAs such as miR-let-7a-5p, miR-24-3p, miR-19-b-1-5p, and miR-450-b-5p to target Extl3-Reg-cyclinD1 pathway and facilitate pancreatic restoration [69]. MSC-EV-delivered miR-146a also reverses diabetic β-cell dedifferentiation and improves β-cell function by inhibiting Numb expression [70]. In addition, recent studies have revealed that MSC-EVs show high efficacy to alleviate peripheral insulin resistance. For instance, Yap et al. demonstrated that MSC-EVs represent promising therapeutic agents to enhance glucose uptake by skeletal muscle cells and ameliorate type 2 DM [71]. Furthermore, the findings that MSC-EVs promote hepatic glycolysis, glycogen storage, and lipolysis, and reduce gluconeogenesis in diabetic rats suggest that MSC-EVs can improve insulin sensitivity in peripheral organs [72]. These findings suggest that MSC-EVs treatment serves as an effective strategy to alleviate DM by regulating the immune microenvironment, recovering β-cell function and mass, and alleviating peripheral insulin resistance (Figure 2).

### 3.2. Diabetic Nephropathy

As one of the severe microvascular complications of DM, diabetic nephropathy is the leading cause of end-stage renal disease worldwide [73]. Diabetic nephropathy is a chronic disease characterized by diffuse and nodular mesangial matrix expansion, glomerular basement membrane thickening, glomerular hyperfiltration, and tubulointerstitial fibrosis [74]. Previous studies have revealed that MSC transplantation shows the ability to directly rescue kidney damage in diabetic conditions [75], whereas cell therapy-induced potential side effects limit its clinical translation. Although autologous cell transplantation can partly solve the problem of immune rejection, hyperglycemia may cause abnormalities of intrinsic MSC in diabetic patients, resulting in insufficient repairing effects. Nagaishi et al. developed a novel approach based on umbilical cord MSC-EVs to improve the renal therapeutic value of injured bone marrow MSC [76]. The proliferation, motility, endoplasmic reticular functions, and EV secretion ability of type 1 and type 2 diabetes-derived bone marrow MSC can be enhanced after umbilical cord MSC-EVs treatment. In addition, Li et al. demonstrated that GW4869 (an inhibitor of EV secretion) administration significantly impairs the antifibrosis effects of MSC, suggesting that EV release serves as the major mechanism responsible for MSC-induced diabetic nephropathy therapy [77]. Therefore, MSC-EV-mediated cell-free strategy has become a new research direction. Many preclinical studies have been adopted to evaluate the role of MSC-EVs in diabetic nephropathy. Wang et al. revealed that STZ-induced diabetic rats after MSC-EV treatment display alleviated pathological changes in renal glomerulus and tubules, and decreased levels of blood glucose, lipids, and viscosity [78]. Moreover, MSC-EVs contain various miRNAs that can prevent the development of renal fibrosis by targeting profibrotic genes in STZ-induced diabetic animal models [79,80]. In db/db mice, MSC-EVs also show the potent capacity to ameliorate hyperglycemia-induced renal apoptosis and epithelial-mesenchymal transition (EMT) by delivering miR-424-5p [81]. Jin and colleagues have identified that miR-486-mediated Smad1 downregulation contributes to the reduction in urine protein, serum creatinine, and blood urea nitrogen in db/db mice treated with MSC-EVs [82]. As one type of terminally differentiated epithelial cells, podocytes exert critical roles in maintaining glomerular filtration barrier function [83]. Hyperglycemia-caused podocyte injury is recognized as the key pathogenic factor of glomerular hyperfiltration and proteinuria during diabetic nephropathy progress [84]. MSC-EVs are reported to alleviate high glucose (HG) conditions-induced EMT of podocytes by delivering miR-215-5p to inhibit zinc finger E-box-binding homeobox 2 (ZEB2) expression [85]. MiR-15b-5p in MSC-EVs is able to cause the downregulation of pyruvate dehydrogenase kinase 4 (PDK4) and VEGF expressions, thus reducing the apoptosis and inflammation of podocytes [86]. Duan et al. revealed the protective role of miR-26a-5p from MSC-EVs to enhance the viability of podocytes by reducing Toll-like receptor 4 (TLR4) expression and inhibiting downstream NF-κB/VEGF pathway [87]. In addition, MSC-EVs also promote the proliferation of human embryonic kidney epithelial cells (HKCs) in HG medium by miR-125b-mediated inhibition on tumor necrosis factor receptor-associated factor 6 (TRAF6) level [88]. Another study by Nagaishi et al. has demonstrated that MSC-EVs not only inhibit tubular epithelial cell apoptosis to maintain the tight junction structure but also reduce the release of inflammatory cytokines [89]. These findings suggest that MSC-EV treatment represents a promising approach to preventing diabetic nephropathy by repairing renal functions and regulating glucose metabolism.

### 3.3. Diabetic Retinopathy

Diabetic retinopathy, a leading cause of vision decline and blindness in adults, is characterized by the loss of retinal cells and the infiltration of inflammatory and oxidative factors, leading to retinal tissue destruction, vascular leakage, and neovascularization [90]. Retinal ischemia is recognized as an important mechanism associated with diabetic retinopathy progress. Mathew et al. demonstrated that the intravitreal injection of MSC-EVs remarkably recovers retinal functions and alleviates neuroinflammation and apoptosis in retinal ischemia models, highlighting the potential of MSC-EVs in the treatment of retinal disorders [91]. In HG medium-cultured human retinal microvascular endothelial cells, MSC-EVs exhibit the ability to inhibit EMT and tube formation [92]. Further studies show that MSC-EV-mediated lncRNA SNHG7 delivery to decrease miR-34a-5p expression is essential for the upregulation of X-box binding protein 1 (XBP1) expression. To determine whether MSC-EVs exert retinal therapeutic roles in vivo, Fu et al. treated STZ-induced diabetic rats with MSC-EVs through intravitreal injection and found that MSC-EVs effectively attenuate retinal structure disruption and vascular injury [93]. Moreover, Ebrahim et al. suggested that the blocking of Wnt/β-catenin signaling explains the MSC-EV-mediated reduction in retinal oxidative stress, inflammation, and angiogenesis [94]. In recent years, researchers have focused on the exploration of functional components within MSC-EVs in diabetic retinopathy therapy. For example, the results of Sun et al. have shown that hyperglycemia-induced upregulation of phosphatase and tensin homolog (PTEN) can inhibit AKT phosphorylation and nuclear factor erythroid 2-related factor 2 (NRF2) expression in retinal tissues, whereas MSC-EVs ameliorate retinal apoptosis and oxidative stress by transporting neuronal precursor cell-expressed developmentally downregulated 4 (NEDD4) to cause PTEN ubiquitination and degradation and activate downstream AKT/NRF2 pathway [95]. The knockdown of NEDD4 significantly impairs the retinal protective potential of MSC-EVs. In addition, the miRNAs in MSC-EVs such as miR-18b, miR-17-3p, miR-486-3p, miR-146a, miR-192, and miR-133b-3p also exert repairing roles to promote the recovery of retinal function and structure [96,97,98,99,100,101]. Furthermore, MSC-EVs are reported to carry brain-derived neurotrophic factor (BDNF) to retinal neurons of diabetic rats and inhibit neuronal apoptosis by activating the TrkB pathway [102]. However, recent evidence has revealed that intravitreally administered MSC-EVs are rapidly absorbed by superficially located cells such as astrocytes, microglia, and retinal neurons, leaving few MSC-EVs to penetrate deeper into the retina [103]. Notably, the retention time of MSC-EVs in retinal cells is less than 14 days. It is necessary to develop novel strategies to maintain MSC-EVs in retinal tissues for achieving increased cell uptake and extended residence time.

### 3.4. Diabetic Wound Healing

A diabetic ulcer is one of the most common complications of DM [104]. A hyperglycemia-induced decline in healing ability makes the diabetic ulcer an important problem threatening the life and health of diabetic patients [105]. The traditional strategy relies on medical treatment and surgical blood flow reconstruction, whereas therapeutic efficacy is not satisfactory [106]. Recently, MSC-EVs have been widely utilized to accelerate diabetic wound healing. This effect is discovered to be closely associated with the property of MSC-EVs to promote wound closure, re-epithelialization, collagen deposition, and neovascularization [107,108,109]. Further studies have demonstrated that MSC-EV-mediated activation of sirtuin 3 (SIRT3)/superoxide dismutase 2 (SOD2) signaling is responsible for the restoration of mitochondrial functions, resulting in the reduction in wound oxidative stress and inflammation [110]. Moreover, Liu et al. revealed that MSC-EVs can enhance the proliferation, migration, and angiogenesis abilities of vascular endothelial cells by promoting hypoxia-inducible factor-1α (HIF-1α) expression in PI3K-AKT-mTOR dependent manner, leading to the acceleration of diabetic wound healing [111]. Accumulating evidence suggests that the abundance of RNAs, especially lncRNAs and miRNAs in MSC-EVs, contributes to wound healing in diabetic animal models. For instance, Li et al. injected MSC-EVs into the skin around the wound of diabetic mice and observed that MSC-EVs prevent the apoptosis and inflammation of fibroblasts to stimulate the wound-healing process by lncRNA H19-induced miR-152-3p inhibition and downstream PTEN upregulation [112]. In addition, MSC-EV-delivered miR-21-5p, miR-17-5p, and miR-146a also exhibit the angiogenic capacity to facilitate diabetic wound healing [113,114,115]. Although MSC-EVs from different sources all exert beneficial functions in skin wound healing, the functional pathways are distinct. Cargo analysis has shown that bioactive molecules in adipose tissue MSC-EVs are highly correlated to neovascularization, and cargos in bone marrow MSC-EVs mainly cause cell proliferation [116]. There is still a great challenge to select the most effective and suitable source of MSC-EVs.

### 3.5. Diabetic Neuropathy

DM is recognized as an essential metabolic risk factor for nerve injury, whereas blood glucose reduction is not enough to prevent neuropathy in patients with type 2 DM [117]. Although the antiepileptic drugs-based strategy is available to alleviate diabetic neuropathy, long-term control remains a challenge due to the complex pathological mechanisms [118]. Recent evidence has shown that MSC-EVs exhibit therapeutic potential to relieve hyperglycemia-induced nerve damage. For instance, MSC-EVs carry various miRNAs that target the TLR4/NF-κB signaling to increase nerve conduction velocity, intraepidermal nerve fibers number, myelin thickness, and axon diameter of sciatic nerves in db/db mice [119]. In addition, Venkat et al. found that MSC-EV-induced decreased miR-9 level enhances the expressions of ATP-binding cassette transporter 1 (ABCA1) and insulin-like growth factor 1 receptor (IGFR1) in the brain, resulting in the elevated density of axon and myelin, improved integrity of blood-brain barrier, and reduced inflammatory response [120]. The tail vein injection of MSC-EVs is also observed to alleviate neuroinflammation in diabetic intracerebral hemorrhage rats by miR-183-5p-mediated inhibition on programmed cell death 4 (PDCD4)/NOD-like receptor family pyrin domain containing 3 (NLRP3) pathway [121]. The diabetic microenvironment-induced damage of neurons and astrocytes in the hippocampus area is considered an important pathological mechanism of cognitive impairment. Nakano et al. demonstrated that MSC-EVs exert similar protective values with MSC to repair injured neurons and astrocytes after intracerebroventricular (ICV) injection [122]. Moreover, the discovery that MSC-EVs recover cognition impairment and histologic abnormity in diabetic mice suggests that MSC-EV administration represents a promising strategy for cognition therapy with application prospects [123]. Lang et al. further revealed that the effects of MSC-EVs to improve the compromised proliferation and neuronal-differentiation ability of hippocampal neural stem cells are closely related to the delivery of miR-21-5p and miR-486-5p (Table 2) [124]. These findings indicate that MSC-EVs can serve as novel candidates for the treatment of diabetic neuropathy (Figure 3).

Although the therapeutic functions of MSC-EVs in DM and diabetic complications have been explored in numerous preclinical studies, the superiority of MSC-EVs in comparison to current treatment approaches such as insulin and hypoglycemic agents should also be investigated. Sun et al. treated STZ-induced diabetic rats with MSC-EVs or insulin, respectively, and found that both insulin and MSC-EVs can reduce blood glucose levels, whereas exogenous insulin injection exerts little effect to improve the individual insulin sensitivity and β-cell injury [66]. By comparison, MSC-EVs display more beneficial functions to increase glucose uptake and metabolism in the liver and muscles and inhibit β-cell apoptosis. Moreover, Yin et al. demonstrated that HG condition-induced apoptosis and dysfunction of podocytes are closely associated with Yes-associated protein (YAP) upregulation, which is reversed by MSC-EV administration [125]. In contrast, single insulin treatment cannot reduce YAP expression and improve podocyte function. These findings reveal that MSC-EVs may exert more effective roles in the treatment of DM and diabetic complications. Currently, one clinical trial involving MSC-EVs for the improvement of β-cell mass in type 1 DM patients has been reported (https://www.clinicaltrials.gov/ct2/show/NCT02138331?term=MSC+exosomes&cond=diabetes&draw=2&rank=1 (accessed on 1 August 2022)). However, the result of this trial is still unknown. More studies should be initiated to promote the clinical application of MSC-EVs in the treatment of DM and diabetic complications.

## 4. Engineered MSC-EVs in the Treatment of DM and Diabetic Complications

Although MSC-EV-mediated cell-free strategy provides considerable promise for DM and diabetic complications therapy, there are also several challenges that may hinder its clinical translation such as heterogeneity and insufficient function. Recent evidence suggests that proper modification can enhance the contents, biodistribution, and bioactivity of MSC-EVs, which may overcome the limitations of natural MSC-EVs. For this purpose, various engineering methods have been established. At present, there are mainly two major strategies including modification of MSC followed by the purification of engineered MSC-EVs and the direct decoration of the isolated MSC-EVs.

### 4.1. Modification of MSC

Gene transfection is a convenient strategy for loading cargo into MSC with the help of viral or plasmid vectors [126]. Through the natural biogenesis process, exogenous cargos can be packaged within MSC-EVs. Recently, this method is widely used to produce engineered MSC-EVs with enhanced therapeutic efficiency for the treatment of hyperglycemia-induced tissue damage. For instance, Wen et al. used EVs from MSC transfected with plasmids encoding shFas and anti-miR-375 to silence Fas and miR-375 in human islets, resulting in the improved viability and function of islets [127]. In an STZ-induced mouse model of diabetic ulcers, EVs are isolated from the lncRNA KFL3-AS1-overexpressing MSC. Compared with natural MSC-EVs, these engineered EVs show elevated abilities to promote blood vessel formation and reduce inflammation by weakening miR-383-induced VEGF downregulation [128]. HOX transcript antisense RNA (HOTAIR) has been reported to play critical roles in mediating the angiogenic effects of endothelial cells by enhancing VEGF expression and reversing miR-761-induced histone deacetylase 1 inhibition [129,130]. Born et al. demonstrated that MSC treated with HOTAIR overexpression plasmids can produce EVs with increased HOTAIR content, which further accelerates angiogenesis and wound healing in db/db mice [131]. Moreover, EVs from mmu_circ_0000250-modified MSC effectively deliver mmu_circ_0000250 to promote SIRT1 expression by adsorbing miR-128-3p, leading to the enhanced neovascularization and decreased apoptosis in wound skin of STZ-induced diabetic mice [132]. In db/db mice, engineered MSC-EVs containing increased miR-146a levels due to the transfection of donor cells display a strengthened therapeutic potential for neurological restoration [133]. Overall, this strategy is simple and feasible but the encapsulation efficiency of MSC still needs further improvement.

Exogenous stimulation represents another efficient strategy to optimize MSC-EVs by adding functional components to co-incubate with MSC. Increasing studies have suggested that MSC treated with chemical or biological factors can enhance the bioactivity of MSC-EVs to further promote wound healing in STZ-induced diabetic rats. As a natural polyphenol compound, resveratrol displays the property to improve endothelial function and promote neovascularization [134]. Hu et al. reported that MSC pretreated with resveratrol increases miR-129 levels in MSC-EVs, resulting in the promotion of diabetic wound healing by inhibiting TRAF6 expression [135]. Another study has also revealed that EVs derived from MSC treated with pioglitazone, a peroxisome proliferator-activated receptor activator, are able to further promote collagen deposition, extracellular matrix remodeling, and VEGF and CD31 expressions through activating the PI3K/AKT/eNOS pathway [136]. Atorvastatin is an inhibitor of 3-hydroxy-3-methylglutaryl-coenzyme A reductase and is widely used to reduce blood lipid in clinics [137]. Recent evidence has indicated that the atorvastatin-induced upregulation of miR-221-3p in MSC-EVs results in the recovery of endothelial cell function, thus alleviating diabetic skin defects [138]. As a hypoxia-mimic compound, deferoxamine can activate a HIF-1α signaling pathway [139]. Compared with natural MSC-EVs, diabetic rats treated with EVs secreted by deferoxamine-stimulated MSC exhibit enhanced angiogenic activity [140]. Moreover, Liu et al. stimulated MSC with melatonin which is a major secretory product of the pineal gland and possesses potent anti-inflammatory ability, and then isolated engineered MSC-EVs, which remarkably attenuate the inflammation response by causing macrophage polarization from M1 to M2 in a diabetic wound [141]. Vascular calcification is characterized by the deposition of calcium phosphate in the cardiovascular structure, which is common in patients with DM. Wang et al. obtained EVs from MSC pretreated with advanced glycation end product bovine serum albumin (AGEs-BSA) and assessed their protective effects on vascular calcification [142]. The results show that engineered MSC-EVs reduce the production of reactive oxygen species and inhibit the AGEs-BSA-induced calcification by delivering increased miR-146a to downregulate thioredoxin-interacting protein (TXNIP) expression. However, the exact mechanism responsible for the functional cargo enrichment in MSC-EVs after exogenous stimulation remains unclear.

The regulatory effects of EVs on recipient cells depend on the state of donor cells. Changing the culture condition of MSC is considered an effective method to enhance the yield and therapeutic value of MSC-EVs. During the culture process, oxygen tension is a key factor affecting the biological functions of MSC. Increasing studies have revealed that a hypoxia microenvironment can activate the differentiation potential of MSC and upregulate stemness gene expressions, which may alter the content of MSC-EVs [143]. Sun et al. collected EVs from tumor necrosis factor-α (TNF-α)-treated MSC cultured in hypoxia conditions, followed by the encapsulation of cationic antimicrobial carbon dots [144]. Treatment with these engineered MSC-EVs significantly promotes neovascularization by stabilizing HIF-1α and inhibits reactive oxygen species generation and inflammation by inducing M2 macrophage polarization, thus accelerating wound healing in STZ-induced diabetic mice. In addition, Ti et al. revealed that lipopolysaccharide (LPS)-preconditioned MSC exhibits elevated paracrine effects to promote the secretion of MSC-EVs, which are able to exert immunotherapeutic effects on wound healing in STZ-induced diabetic rats by transferring let-7b [145]. Static adherent culture mode, known as two-dimensional (2D) culture, is usually adopted for the expansion of MSC. However, due to the reduced cell-to-cell and cell-to-extracellular matrix interactions in 2D culture conditions compared with a three-dimensional (3D) environment in vivo, the original morphology, structure, and function of MSC may be significantly changed, leading to the limited release of MSC-EVs with therapeutic value [146]. Recently, various 3D culture technologies have been developed to improve the yield and repair the efficiency of MSC-EVs, mainly including scaffold-free systems and scaffold-based systems. Currently, there are several scaffold-free 3D culture approaches such as agitated culture condition [147], suspension culture [148], and 3D spherical spatial boundary condition [149]. On the other hand, scaffold-based 3D culture methods are relatively diverse, including hydrogel-assisted 3D culture [150], fibrous scaffold [151], extracellular matrix bioscaffold [152], and 3D microcarrier culture system [153]. MSC shows elevated proliferation and differentiation potential in 3D culture conditions [154]. Cao et al. demonstrated that the MSC-EVs production of the hollow fiber bioreactor-based 3D culture system is 19.4 times higher than 2D culture [155]. Increasingly, studies have revealed that EVs from 3D system-cultured MSC exhibit enhanced therapeutic value in many diseases such as Alzheimer’s disease and traumatic brain injury [156,157]. Future studies should evaluate whether these engineered MSC-EVs can further alleviate DM and diabetic complications. These findings highlight the strengthened repairing function of EVs derived from MSC cultured in improved conditions.

### 4.2. Direct Modification of Isolated MSC-EVs

Although manipulating MSC to obtain engineered MSC-EVs can preserve their structure and most biophysical characteristics, this strategy may interfere with the biogenesis process of EVs and change their several biological functions, leading to unforeseen consequences. The approach based on the modification of isolated MSC-EVs displays the superiority of directly endowing them with desired biological properties on demand. In contrast, methods to directly modify MSC-EVs are generally more efficient.

Hydrogels can ensure the retention of encapsulated cargo at the target tissue and control their release over a prolonged period of time. The introduction of MSC-EVs into hydrogels holds immense promise as a cell-free sustained delivery platform in the treatment of DM and diabetic complications. Current studies involving this engineering strategy mainly focus on its application in diabetic wound healing. Zhang et al. developed a bioactive scaffold containing polyvinyl alcohol/alginate nanohydrogel loaded with MSC-EVs, which significantly upregulate VEGF expression and accelerate the wound healing process in diabetic rats [158]. Wang et al. reported that MSC-EV-encapsulated FHE hydrogel further enhances the therapeutic efficiency in diabetic full-thickness cutaneous wounds, characterized by an increased wound closure rate, angiogenesis, re-epithelization, and collagen deposition compared with MSC-EVs or FHE hydrogel treatment [159]. Moreover, MSC-EVs binding to the porcine small intestinal submucosa-based hydrogel via peptides achieve the sustained release of bioactive molecules to improve the biological functions of fibroblasts in STZ-induced diabetic models [160]. Another study by Yang et al. also demonstrated that Pluronic F-127 hydrogel modification results in the elevated ability of MSC-EVs to promote the expressions of VEGF and Ki67 and the regeneration of granulation tissue [161]. Chitosan-based multifunctional composite hydrogel represents an ideal dressing for wound healing due to its great biocompatibility, biodegradability, and security. Many attempts have been made to use this polymer material to modify MSC-EVs for enhancing their repairing potential. For instance, Geng et al. utilized carboxyethyl chitosan-dialdehyde carboxymethyl cellulose hydrogel to package MSC-EVs, which show the property to synergistically inhibit inflammation response and promote neovascularization in wound area [162]. In addition, diabetic rats treated with MSC-EV-loaded chitosan/silk hydrogel have more neo-epithelium, collagen, and vessels compared with control groups [163]. The research on the combination of cell manipulation and EV modification to decorate MSC-EVs has also achieved encouraging progress. Tao et al. isolated EVs from miR-126-3p-overexpressing MSC and then constructed a controlled-release system after the adjunction of chitosan hydrogel [164]. These engineered MSC-EVs significantly stimulate re-epithelialization, angiogenesis, and collagen maturity, thus providing a new approach to diabetic ulcer therapy (Table 3).

Elevating the content of bioactive molecules in MSC-EVs serves as another strategy to improve their therapeutic potential. Various methods have been established to directly load exogenous cargos into isolated MSC-EVs including incubation, electroporation, sonication, extrusion, and freeze and thaw cycles (Table 4). Several hydrophobic drugs can be encapsulated into MSC-EVs after simple incubation through the interaction with the hydrophobic membrane of MSC-EVs [165]. Electroporation relies on the stimulation of electrical fields to create transit pores on the membrane of MSC-EVs, leading to the introduction of bioactive molecules and drugs [166]. This method displays high loading efficiency, whereas the aggregation of MSC-EVs may affect their integrity and protective effects. Sonication-induced shear forces result in the deformation of the EV membrane, thus allowing the loading of exogenous cargo into MSC-EVs [167]. The setting of sonication parameters determines the loading efficiency. Violent sonication may cause the destruction of MSC-EVs. Repeated extrusion provides opportunities to change the membrane structure of MSC-EVs for the entry of cargo [168]. This strategy shows the advantage of efficient packaging, whereas the membrane properties of MSC-EVs may be irreversibly injured. Similarly, the freeze-thaw method promotes cargo encapsulation by ice crystal-induced changes in the membrane of MSC-EVs, which may impair the integrity of EVs and bring safety risks [169]. Although these modification approaches have been widely utilized to enhance the therapeutic value of MSC-EVs in various diseases, there are few studies on the application of engineered MSC-EVs after direct cargo loading for the therapy of DM and diabetic complications. Gondaliya et al. prepared engineered MSC-EVs encapsulated with an miR-155 inhibitor through a CaCl_2_-modified co-incubation method [170]. Further studies demonstrate that miR-155 inhibitor-loaded MSC-EVs show an enhanced ability to promote collagen deposition, neovascularization, and re-epithelialization to promote wound healing in STZ-induced diabetic mice. The construction of engineered MSC-EVs through loading exogenous therapeutic agents to prevent hyperglycemia-induced tissue injury still needs further investigation (Figure 4).

## 5. Challenges and Perspectives

MSC-EVs have shown great value in regenerative medicine due to their enriched bioactive molecules and high biocompatibility and stability. Although substantial breakthroughs have been made in the field of MSC-EV-mediated cell-free therapy for DM and diabetic complications, there still exist many challenges that may hinder the clinical application of MSC-EVs: (1) The large-scale production of high-quality MSC-EVs is the major challenge currently. The traditional isolation methods such as ultracentrifugation have several disadvantages including low yield and time-consuming processes, which cannot meet the clinical requirement [171]. There is an urgent need to develop new separation approaches to rapidly obtain MSC-EVs with high purity. In addition, recent studies have demonstrated that improved culture conditions, especially the 3D culture system, can remarkably enhance the production of MSC-EVs, whereas the production costs and technical difficulties have not been completely solved [172]. Accurate characterization of MSC-EVs to ensure their quality is critically important for their utilization in tissue regeneration. There is still a lack of standardization to evaluate the heterogeneity, components, and structure of MSC-EVs. (2) The dosage and administration route of MSC-EVs require further investigations. Future studies should focus on the exploration of the best intervention way for specific diseases. (3) The modification of MSC-EVs represents a promising strategy to elevate their therapeutic value for hyperglycemia-induced tissue injury. However, current engineering strategies may affect the structure and biological functions of MSC-EVs [173]. The optimized methods for cargo loading into MSC-EVs need further development to realize the maximum encapsulation efficiency and maintain the inherent features of MSC-EVs. (4) Illustrating the intracellular fate of natural MSC-EVs or engineered MSC-EVs after administration is critical for therapeutic applications, including absorption, clearance, and biodistribution. Comprehensive preclinical studies should be conducted to evaluate the biosafety of MSC-EVs. In addition, as a result of the changed compositions of engineered MSC-EVs after modification, long-term safety examinations in vivo should be addressed.

## 6. Conclusions

Overall, the MSC-EV-mediated cell-free strategy is attractive and promising in the treatment of DM and diabetic complications. Current clinical treatment for DM mainly relies on the injection of insulin and some hypoglycemic agents. However, this strategy can only temporarily control blood glucose and even cause side effects such as subcutaneous nodules, diarrhea, and obesity [174]. Moreover, exogenous insulin administration may aggravate β-cell dysfunction, leading to the development of peripheral insulin resistance and diabetic complications [175]. Various preclinical studies have demonstrated that MSC-EVs show effective roles in alleviating hyperglycemia in diabetic animal models by improving β-cell mass, promoting insulin sensitivity, and increasing glucose uptake and metabolism in peripheral tissues. In addition, MSC-EVs also represent promising candidates to ameliorate diabetic complications through multiple mechanisms. The application of MSC-EVs brings a key breakthrough in the treatment of DM and diabetic complications. With the development of modification technologies, the construction of engineered MSC-EVs represents a new research direction. Attempts to modify MSC-EVs for improving their protective effects are still in their infancy. There is much room for exploration of the therapeutic potential of MSC-EVs, especially engineered MSC-EVs, in hyperglycemia-induced tissue injury.

## Figures and Tables

**Figure 1 pharmaceutics-14-02208-f001:**
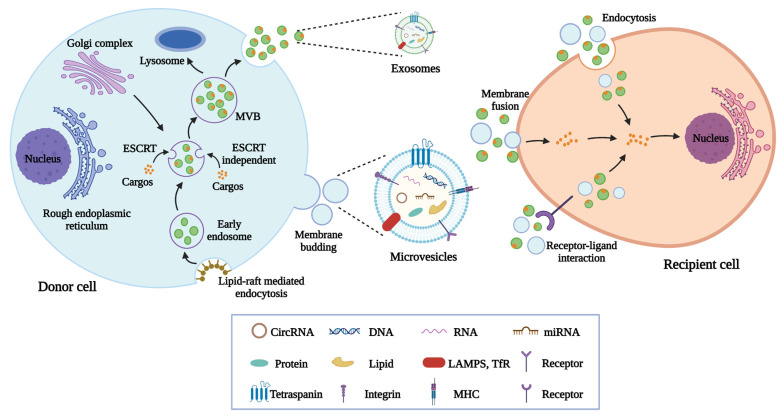
The biogenesis, release, and internalization of EVs. The generation of exosomes involves several processes including endocytosis, endosomes and MVBs formation, and the fusion of MVBs with plasma membranes. Microvesicles are formed after the outward budding of plasma membranes. EVs transport cargos including proteins, nucleic acids, and lipids to recipient cells through endocytosis, membrane fusion, and receptor-ligand interaction.

**Figure 2 pharmaceutics-14-02208-f002:**
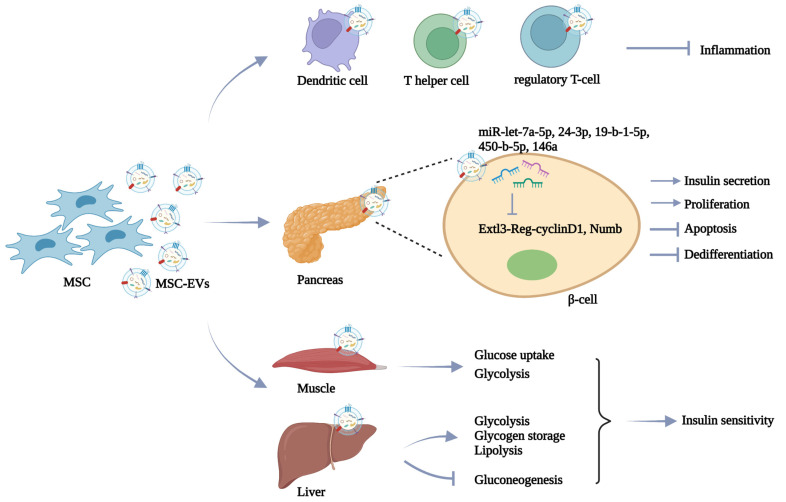
MSC-EVs in DM therapy. MSC-EVs display the ability to alleviate DM progress by regulating immune response, relieving β-cell damage, and reversing peripheral insulin resistance.

**Figure 3 pharmaceutics-14-02208-f003:**
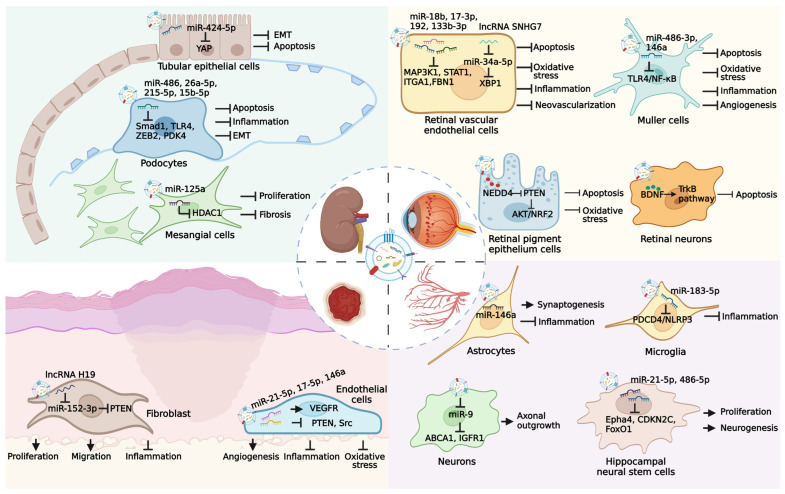
Therapeutic potential of MSC-EVs in several major diabetic complications. Long-term hyperglycemia microenvironment can cause the occurrence of many diabetic complications, such as diabetic nephropathy, diabetic retinopathy, diabetic ulcer, and diabetic neuropathy. MSC-EVs carry various bioactive molecules that exert effective roles in the treatment of diabetic complications through multiple pathways.

**Figure 4 pharmaceutics-14-02208-f004:**
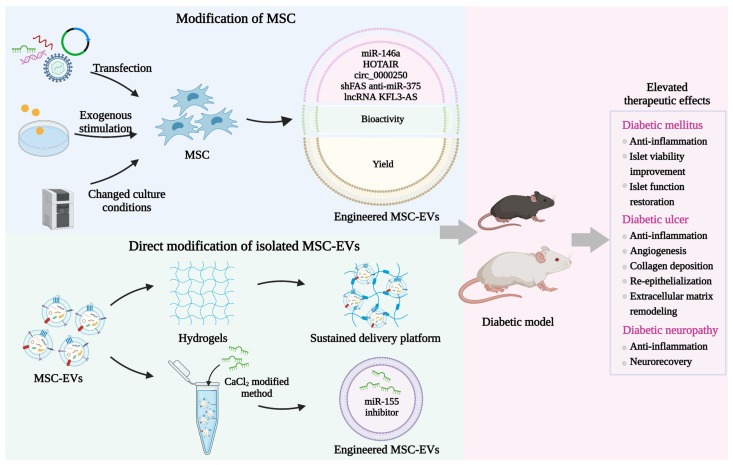
Engineered MSC-EVs in the treatment of DM and diabetic complications. Various engineering strategies have been established to amplify the protective functions of MSC-EVs by loading exogenous cargos and increasing their bioactivity, yield, and retention. Manipulating MSC and direct modification of isolated MSC-EVs are two major methods to construct engineered MSC-EVs, which have shown elevated therapeutic effects in the treatment of DM, diabetic ulcers, and diabetic neuropathy currently.

**Table 1 pharmaceutics-14-02208-t001:** Evaluation of isolation methods of MSC-EVs.

Method	Principle	Advantages	Limitations	Reference
Ultracentrifugation	According to the size, density, and shape of MSC-EVs	Low cost, simple operation, suitable for large samples	Time-consuming, low yield, poor integrity	[47]
Density gradient centrifugation	Based on the density of MSC-EVs	High purity	Time consuming, complex operation	[48]
Size exclusion chromatography	Based on the size of MSC-EVs	Simple operation, high yield, high purity, good integrity	High cost, suitable for low sample volume	[49]
Immunoaffinity capture	Specific binding of antibody to the surface marker of MSC-EVs	High purity	High cost, low yield	[50]
Ultrafiltration	Based on the size of MSC-EVs	Efficiency, simple operation	Low purity	[51]
Polymer precipitation	Changing the solubility and dispersibility of MSC-EVs	High yield, simple operation	Low purity, high cost	[52]

**Table 2 pharmaceutics-14-02208-t002:** Therapeutic function of natural MSC-EVs in DM and diabetic complications.

Disease	Animal Model	Injection of MSC-EVs	Effect of MSC-EVs In Vivo	Cell Culture	Effect of MSC-EVs In Vitro	Reference
DM	N/A	N/A	N/A	Dendritic cells from diabetic patients	Induce immature IL-10- secreting dendritic cells to alleviate inflammation	[63]
DM	NOD mice	Intravenous	Inhibit the activation of antigen- presenting cells and the development of T helper cells	N/A	N/A	[64]
DM	STZ-induced diabetic mice	Intraperitoneal	Enhance the islet number and improve glycemic control by increasing regulatory T-cell population	N/A	N/A	[65]
DM	STZ-induced diabetic rats	Intravenous	Reduce blood glucose level, inhibit β-cell apoptosis, and alleviate peripheral insulin resistance	PA-treated LO2 cells and HG medium-treated L6 cells	Enhance glucose uptake and glycolysis in L6 cells, and reduce glycogenolysis in LO2 cells	[66]
DM	NOD mice injected with STZ	Intrapancreatic	Increase islet number and β-cell mass, elevate insulin level, and reduce blood glucose	HMVECs cultured in serum-starved conditions	Enhance the tube formation ability of HMVECs	[67]
DM	STZ-induced diabetic rats	Intravenous	Promote islet regeneration and insulin production through Pdx-1 mechanism	N/A	N/A	[68]
DM	STZ-induced diabetic mice	Intravenous	Alleviate hyperglycemia and facilitate pancreatic regeneration by regulating the Extl3-Reg-cyclinD1 pathway	N/A	N/A	[69]
DM	STZ-induced diabetic rats	Intravenous	Improve β-cell function by miR-146a-mediated inhibition on Numb expression	INS-1 cells cultured in HG medium	Alleviate cell dedifferentiation	[70]
DM	STZ-induced diabetic rats	Intravenous	Ameliorate insulin resistance and relieve the structural injury of pancreas, kidney, and liver	HSkMCs cultured in serum-starved conditions	Increase glucose uptake by HSkMCs	[71]
DM	STZ-induced diabetic rats	Intravenous	Promote hepatic glycolysis, glycogen storage and lipolysis, and reduce gluconeogenesis	PA-treated LO2 cells	Promote glycolysis and glycogen synthesis, and inhibit gluconeogenesis	[72]
Diabetic nephropathy	STZ-induced diabetic rats	Intravenous	Alleviate pathologic changes in the renal glomerulus and tubule	N/A	N/A	[78]
Diabetic nephropathy	STZ-induced diabetic rats	Intravenous	Suppress mesangial hyperplasia and kidney fibrosis by miR-125a-induced HDAC1 inhibition	Glomerular mesangial cells cultured in HG medium	Reduce IL-6, collagen I, and fibronectin expressions, and promote cell apoptosis	[79]
Diabetic nephropathy	STZ-induced diabetic mice	Intravenous	Suppress renal fibrosis	N/A	N/A	[80]
Diabetic nephropathy	Db/db mice	Intravenous	Alleviate renal apoptosis and EMT by miR-424-5p-induced YAP1 inhibition	HK2 cells cultured in HG medium	Promote cell proliferation and inhibit EMT process	[81]
Diabetic nephropathy	Db/db mice	Intravenous	Improve renal function and inhibit renal apoptosis by miR-486-induced Smad1 inhibition	MPC5 cells cultured in HG medium	Improve cell viability and inhibit cell apoptosis	[82]
Diabetic nephropathy	N/A	N/A	N/A	MPC5 cells cultured in HG medium	Alleviate EMT by miR-215-5p-induced ZEB2 inhibition	[85]
Diabetic nephropathy	N/A	N/A	N/A	MPC5 cells cultured in HG medium	Alleviate inflammation and apoptosis by miR-15b-5p-induced PDK4 inhibition	[86]
Diabetic nephropathy	Db/db mice	Intravenous	Improve renal function and inhibit renal apoptosis by miR-26a-5p- induced TLR4 inhibition	MPC5 cells cultured in HG medium	Promote cell proliferation	[87]
Diabetic nephropathy	N/A	N/A	N/A	HKCs cultured in HG medium	Promote cell proliferation by miR-125b-induced TRAF6 inhibition	[88]
Diabetic nephropathy	STZ-induced diabetic mice	Intravenous	Inhibit renal apoptosis, inflammation, and fibrosis.	N/A	N/A	[89]
Diabetic retinopathy	N/A	N/A	N/A	Human RMECs cultured in HG medium	Suppress EMT and tube formation by delivering lncRNA SNHG7	[92]
Diabetic retinopathy	STZ-induced diabetic rats	Intravitreal	Alleviate retinal structure disruption and vascular damage	N/A	N/A	[93]
Diabetic retinopathy	STZ-induced diabetic rats	Intravitreal	Inhibit retinal oxidative stress, inflammation, and angiogenesis by suppressing Wnt/β-catenin signaling	N/A	N/A	[94]
Diabetic retinopathy	STZ-induced diabetic rats	Intravitreal	Alleviate retinal apoptosis and oxidative stress by NEDD4-mediated PTEN inhibition	RPE cells cultured in HG medium	Inhibit cell apoptosis and oxidative injury	[95]
Diabetic retinopathy	STZ-induced diabetic rats	Intravitreal	Ameliorate retinal vascular leakage and inflammation by miR-18b-induced MAP3K1 inhibition	Human RMECs cultured in HG medium	Inhibit cell inflammation and apoptosis	[96]
Diabetic retinopathy	STZ-induced diabetic mice	Intravitreal	Alleviate retinal inflammation and oxidative stress by miR-17-3p-induced STAT1 inhibition	N/A	N/A	[97]
Diabetic retinopathy	STZ-induced diabetic mice	Intravitreal	Alleviate retinal apoptosis, inflammation, and oxidative stress by miR-486-3p-induced TLR4 inhibition	Muller cells cultured in HG medium	Promote cell proliferation	[98]
Diabetic retinopathy	STZ-induced diabetic mice	Intravitreal	Suppress retinal inflammation by miR-146a-induced TLR4 inhibition	N/A	N/A	[99]
Diabetic retinopathy	STZ-induced diabetic rats	Intravitreal	Ameliorate retinal inflammation and angiogenesis by miR-192-induced ITGA1 inhibition	HG medium-cultured RPE cells, Muller cells and human RMECs	Inhibit RPE cell apoptosis, Muller cell activation and RMECs proliferation	[100]
Diabetic retinopathy	KK/Upj-Ay mice	Intravitreal	Ameliorate retinal oxidative stress and neovascularization by miR-133b-3p- induced fibrillin-1 inhibition	HG medium-cultured mouse RMECs	Alleviate cell oxidative stress and angiogenesis	[101]
Diabetic retinopathy	N/A	N/A	N/A	Rat retinal neurons cultured in HG medium	Alleviate neuronal apoptosis by activating BDNF-TrkB pathway	[102]
Diabetic wound healing	STZ-induced diabetic mice	At the wound site	Promote granulation tissue formation and angiogenesis	HUVECs cultured in HG medium	Promote cell proliferation and tube formation, and inhibit oxidative stress and inflammation	[108]
Diabetic wound healing	Db/db mice	At the wound site	Promote wound closure, re-epithelialization, and collagen synthesis	Human dermal fibroblasts cultured in HG medium	Enhance cell proliferation and migration	[109]
Diabetic wound healing	Db/db mice	Subcutaneous	Promote angiogenesis and wound closure by activating SIRT3/SOD2 signaling	HUVECs cultured in HG medium	Promote cell proliferation, migration, and tube formation	[110]
Diabetic wound healing	STZ-induced diabetic rats	Intradermal	Accelerate wound closure, collagen deposition, and angiogenesis	AGE-treated HUVECs	Promote cell proliferation, migration, and tube formation	[111]
Diabetic wound healing	STZ-induced diabetic mice	Injected into the skin around the wound	Promote angiogenesis and collagen deposition, and inhibit inflammation by delivering lncRNA H19	Fibroblast cultured in HG medium	Promote cell proliferation and migration	[112]
Diabetic wound healing	STZ-induced diabetic rats	Intramuscular	Promote blood perfusion and angiogenesis by delivering miR-21-5p	HUVECs cultured in HG medium	Promote cell proliferation, migration, and tube formation	[113]
Diabetic wound healing	Db/db mice	Injected into the skin around the wound	Promote angiogenesis by miR-17-5p-induced PTEN inhibition	HUVECs cultured in HG medium	Promote cell proliferation, migration, and tube formation	[114]
Diabetic wound healing	STZ-induced diabetic mice	Injected into the skin around the wound	Accelerate wound closure and angiogenesis by miR-146a-induced Src inhibition	HUVECs cultured in HG medium	Promote cell proliferation, migration, and tube formation	[115]
Diabetic neuropathy	Db/db mice	Intravenous	Increase nerve conduction velocity, intraepidermal nerve fiber number, and myelin thickness	N/A	N/A	[119]
Diabetic neuropathy	STZ-induced diabetic rats	Intravenous	Improve blood brain barrier integrity, promote white matter remodeling, and inhibit inflammation	N/A	N/A	[120]
Diabetic neuropathy	Db/db mice	Intravenous	Alleviate neuroinflammation by miR-183-5p-induced inhibition on PDCD4/NLRP3 signaling	BV2 cells cultured in HG medium	Inhibit cell oxidative stress, inflammation, and apoptosis	[121]
Diabetic neuropathy	STZ-induced diabetic mice	ICV	Inhibit oxidative stress, increase synaptic density, and repair damaged neurons and astrocytes	N/A	N/A	[122]
Diabetic neuropathy	STZ-induced diabetic mice	Intracranial	Improve cognitive impairment and histological abnormalities	N/A	N/A	[123]
Diabetic neuropathy	STZ-induced diabetic mice	Intravenous	Recover proliferation and neuronal-differentiation capacity of hippocampal neural stem cells	N/A	N/A	[124]

Abbreviations: HMVECs: human microvascular endothelial cells; RMECs: retinal microvascular endothelial cells; HSkMCs: human skeletal muscle cells; HUVECs: human umbilical vein endothelial cells; ITGA1: integrin subunit α1; MAP3K1: mitogen-activated protein kinase kinase kinase 1; MPC5: mouse podocyte clone 5; N/A: not applicable; PA: palmitic acid; RPE: retinal pigment epithelium; STAT1: signal transducer and activator of transcription 1.

**Table 3 pharmaceutics-14-02208-t003:** Hydrogel-modified MSC-EVs for diabetic wound healing.

EVs Source	Hydrogel Details	Hydrogel Loading	Diabetic Model	Outcomes	Reference
Umbilical cord-derived MSC	polyvinyl alcohol/alginate nanohydrogel	Mixing, stirring, and gelation	STZ-induced diabetic rats	Increased angiogenesis by promoting VEGF expression	[158]
Adipose-derived MSC	FHE hydrogel composed of Pluronic F127, oxidative hyaluronic acid, and poly-ε-lysine	Mixing, stirring, and gelation	STZ-induced diabetic mice	Enhanced wound closure rates, angiogenesis, re-epithelization, and collagen deposition	[159]
Umbilical cord-derived MSC	Porcine small intestinal submucosa-based hydrogel	Fusion peptide- mediated binding of MSC-EVs to hydrogel	STZ-induced diabetic rats	Elevated granulation tissue and collagen fiber formation and neovascularization	[160]
Umbilical cord-derived MSC	Pluronic F-127 hydrogel	Mixing and gelation	STZ-induced diabetic rats	Enhanced wound closure rate and granulation tissue regeneration by promoting CD31, Ki67 and VEGF expressions	[161]
Bone marrow MSC	Carboxyethyl chitosan-dialdehyde carboxymethyl cellulose hydrogel	Mixing, stirring, and gelation	STZ-induced diabetic rats	Increased angiogenesis and reduced inflammation	[162]
Gingival MSC	Chitosan/silk hydrogel sponge	Seeded on the hydrogel sponge	STZ-induced diabetic rats	Enhanced re-epithelialization, collagen deposition, angiogenesis, and neuronal ingrowth	[163]
Synovium MSC	Chitosan hydrogel	Mixing, stirring, and gelation	STZ-induced diabetic rats	Accelerated re-epithelialization, angiogenesis, and collagen maturity	[164]

**Table 4 pharmaceutics-14-02208-t004:** Evaluation of strategies for cargo loading into MSC-EVs.

Strategy	Method	Principle	Advantages	Limitations	Reference
Modification of MSC	Transfection	Plasmids or virus-mediated cargos delivery	Simple; Maintain the integrity of MSC-EVs	Cytotoxicity; low specificity and efficiency	[126]
Direct modification of MSC-EVs	Incubation	Interaction between cargos and the membrane of MSC-EVs	Simple and feasible	Low loading efficiency	[165]
Direct modification of MSC-EVs	Electroporation	Transient voltage-induced the generation of pores on the membrane	Rapid; High loading efficiency	Aggregation of MSC-EVs; may impair the integrity of MSC-EVs	[166]
Direct modification of MSC-EVs	Sonication	Membrane deformation	High loading efficiency	Aggregation of MSC-EVs; may impair the integrity of MSC-EVs	[167]
Direct modification of MSC-EVs	Extrusion	Mechanical force-induced the temporary destruction of the membrane	Efficient packaging	May change the membrane property	[168]
Direct modification of MSC-EVs	Freeze–thaw cycles	Ice crystals-induced the temporary destruction of the membrane	High loading efficiency	Aggregation of MSC-EVs; may change the membrane structure	[169]

## Data Availability

Not applicable.

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
