# Peer review of "Mesenchymal Stem Cell-Derived Extracellular Vesicles: A Potential Therapy for Diabetes Mellitus and Diabetic Complications"

_pharmaceutics, 2022, doi:10.3390/pharmaceutics14102208_

Round 1

Reviewer 1 Report

The review by Sun F et al. reviews the potential role of MSC-EVs for the therapy of Diabetes and diabetic complications. This scientific work is very interesting,  nicely written and contains well-organized information. It seems informative and well-explained to the reader, in particular the images are well done and allow an excellent synthesis of the biological effects. 

I appreciate the authors' efforts in providing this comprehensive review, which will be more helpful to the scholars working in the area. However, it may be taken into consideration for publication after the following changes are made. I would suggest the following corrections:

Major point

1. In the title the english form of "...diabetic complicationtherapy" is not correct. I would suggest to remove "therapy" from the title or changing the title to "Mesenchymal Stem Cell-Derived Extracellular Vesicles: A Potential Therapy for Diabetes and Diabetic Complications". 

2. There was no method used in this review to collect the data, please add the method used at least in the abstract (e.g. "PubMed library was searched from ... to ..., using a combination of Medical Subject Headings (MeSH) and keywords related to ...")

3. In section 3 and 4 the authors have covered several studies, however, the experimental model has not always been specified. The authors need to clearly delineate studies carried out in vitro in cell culture from those in animals in vivo or in human patients (for example at the beginning of each paragraph, or by adding a table with the list of studies carried out, making the model explicit).

Minor point

1. Add the list of abbreviations used in main text at the end of the article

2. Mesenchimal stem cell-derived EVs should be abbreviated to MSC-derived EVs instead of MSCs-derived EVs. This abbreviation should be corrected in the title, abstract, main text and figures

Author Response

Oct 10, 2022

Pharmaceutics

Manuscript ID: pharmaceutics-1951709

Title: Mesenchymal Stem Cell-Derived Extracellular Vesicles: A Potential Therapy for Diabetes and Diabetic Complications

Authors: Fengtian Sun, Yuntong Sun, Feng Wu, Wenrong Xu *, Hui Qian *

Dear Editor,

Thank you for your e-mail dated Oct 2, 2022 regarding the review of our manuscript. We appreciate your assessments of the manuscript and have found that the comments and suggestions are helpful in preparation of the revised manuscript. We revised the manuscript as suggested by reviewers. In the revised manuscript, all the amendments were highlighted by yellow words. The following are our point-by-point responses, in order of the comments about the manuscript:

Q: In the title the english form of "...diabetic complications therapy" is not correct. I would suggest to remove "therapy" from the title or changing the title to "Mesenchymal Stem Cell-Derived Extracellular Vesicles: A Potential Therapy for Diabetes and Diabetic Complications".

Response: Thanks for your suggestion. We have changed the title to “Mesenchymal Stem Cell-Derived Extracellular Vesicles: A Potential Therapy for Diabetes and Diabetic Complications” in our revised manuscript.

Q: There was no method used in this review to collect the data, please add the method used at least in the abstract (e.g. "PubMed library was searched from ... to ..., using a combination of Medical Subject Headings (MeSH) and keywords related to ...").

Response: Thanks for your suggestion. We have added the method used in the abstract of the revised manuscript. “In this review, the PubMed library was searched from inception to August 2022, using a combination of Medical Subject Headings (MeSH) and keywords related to MSC-EVs, DM, and diabetic complications.”

Q: In section 3 and 4 the authors have covered several studies, however, the experimental model has not always been specified. The authors need to clearly delineate studies carried out in vitro in cell culture from those in animals in vivo or in human patients (for example at the beginning of each paragraph, or by adding a table with the list of studies carried out, making the model explicit).

Response: Thanks for your suggestion. We have added a table named Table 2 with the list of studies carried out to make the experimental model explicit in section 3. And we have also clarified the experimental model of the mentioned studies in section 4 and added a table named Table 3 in section 4.

Q: Add the list of abbreviations used in main text at the end of the article.

Response: Thanks for your suggestion. We have added the list of abbreviations used in main text at the end of the revised manuscript.

Q: Mesenchimal stem cell-derived EVs should be abbreviated to MSC-derived EVs instead of MSCs-derived EVs. This abbreviation should be corrected in the title, abstract, main text and figures.

Response: Thanks for your correction. We have corrected this abbreviation in the title, abstract, main text and figures of the revised manuscript.

We hope that the above responses meet the expectations of the reviewer. The comments and suggestions are helpful in improving the quality of our manuscript.

Thank you for your consideration. We look forward to hearing from you and to publication of this manuscript.

Sincerely,

Hui Qian, Prof.

School of Medicine, Jiangsu University,

301 Xuefu Road, 212013, Zhenjiang, Jiangsu, P.R. China.

E-mail: lstmmmlst@163.com

Reviewer 2 Report

In the Manuscript by Sun et al., the authors made a review of literature data on the use of mesenchymal stem cells-derived extracellular vesicles (MSCs-EVs) as possible new therapeutic tools in the treatment of diabetes mellitus (DM). They illustrated the characteristic and the biogenesis of the three main classes of EVs, then they focues their attention on the role of MSCs-EVs in the treatment of DM-related complications, in particular Diabetic nephropathy, retinopathy, wound healing and neuropathy. Finally, they reported the advances in the EVs engineering as a future research field to improve MSCs-EVs yield and biological activity.

The review is well conceived, structured, and written ad it is easily readable even for a non-specialized reader; the sections are complete and give a good landscape of the literature data for the relevant topic.

Collectively, I consider the MS by Sun et al., suitable for publication in Pharmaceutics in the present form.

Author Response

Oct 10, 2022

Pharmaceutics

Manuscript ID: pharmaceutics-1951709

Title: Mesenchymal Stem Cell-Derived Extracellular Vesicles: A Potential Therapy for Diabetes and Diabetic Complications

Authors: Fengtian Sun, Yuntong Sun, Feng Wu, Wenrong Xu *, Hui Qian *

Dear Editor,

Thank you for your e-mail dated Oct 2, 2022 regarding the review of our manuscript. We appreciate your assessments of the manuscript and have found that the comments and suggestions are helpful in preparation of the revised manuscript. We revised the manuscript as suggested by reviewers. In the revised manuscript, all the amendments were highlighted by yellow words. The following are our point-by-point responses, in order of the comments about the manuscript:

In the Manuscript by Sun et al., the authors made a review of literature data on the use of mesenchymal stem cells-derived extracellular vesicles (MSCs-EVs) as possible new therapeutic tools in the treatment of diabetes mellitus (DM). They illustrated the characteristic and the biogenesis of the three main classes of EVs, then they focues their attention on the role of MSCs-EVs in the treatment of DM-related complications, in particular Diabetic nephropathy, retinopathy, wound healing and neuropathy. Finally, they reported the advances in the EVs engineering as a future research field to improve MSCs-EVs yield and biological activity.

The review is well conceived, structured, and written ad it is easily readable even for a non-specialized reader; the sections are complete and give a good landscape of the literature data for the relevant topic.

Collectively, I consider the MS by Sun et al., suitable for publication in Pharmaceutics in the present form.

Response: Thanks for your positive comments. We appreciate your assessments of the manuscript.

Sincerely,

Hui Qian, Prof.

School of Medicine, Jiangsu University,

301 Xuefu Road, 212013, Zhenjiang, Jiangsu, P.R. China.

E-mail: lstmmmlst@163.com

Reviewer 3 Report

I have read with interest the paper entitled “Mesenchymal Stem Cells-Derived Extracellular Vesicles: An Emerging Frontier for Diabetes and Diabetic Complications Therapy”. The article is well structured and the general idea appears attractive. Furthermore, the use of MSC-derived extracellular vesicles for the treatment of diabetes represents a novelty in the field. However, there are many points that should be addressed before the publication of this article:

MAJOR POINTS:

1.    Authors should clarify which is the clinical benefit of MSCs-EVs therapy for Diabetes Mellitus and their clinical complications in comparison to current therapies such as therapeutic insulin and hypoglycemic agents. They could specify the percentages of clinical improvement to assess the real benefit of this treatment.

2.    For the third section it could be interesting to organize the different preclinical studies to improve understanding of the text. Apparently, it makes more sense to explain the preclinical studies based on similar results obtained in order to appreciate the major benefits.

MINOR POINTS:

1.    In lines, 52-54, as well as in the section 2, authors mention the classification of EVs, however, I would recommend authors to follow the 2018 MISEV guidelines of the ISEV and provide a more rigorous classification. Authors should start from Small and Large EVs and based on the markers and pathways followed continue with an in depth classification.

2.    In lines 96-98 we recommend mentioning the role of lipids in the biogenesis process as it appears in figure 1 for a better comprehension of the text.

3.    The explanation of figure 1 could appear between lines 128-129 in order to enhance a correct follow-up of the text.

4.    In lines 72 and 182 we have detected two orthographic mistakes. Before and is not necessary to include a coma and the name of Sun et al., should be in capital letters.

5.    In line 203 appears the biological molecule GW4869 please include an explanation for better comprehension.

6.    The conclusion could be extended by explaining the clinical breakthrough that MSCs-EV could bring in the Diabetes Mellitus field.   

Author Response

Oct 10, 2022

Pharmaceutics

Manuscript ID: pharmaceutics-1951709

Title: Mesenchymal Stem Cell-Derived Extracellular Vesicles: A Potential Therapy for Diabetes and Diabetic Complications

Authors: Fengtian Sun, Yuntong Sun, Feng Wu, Wenrong Xu *, Hui Qian *

Dear Editor,

Thank you for your e-mail dated Oct 2, 2022 regarding the review of our manuscript. We appreciate your assessments of the manuscript and have found that the comments and suggestions are helpful in preparation of the revised manuscript. We revised the manuscript as suggested by reviewers. In the revised manuscript, all the amendments were highlighted by yellow words. The following are our point-by-point responses, in order of the comments about the manuscript:

I have read with interest the paper entitled “Mesenchymal Stem Cells-Derived Extracellular Vesicles: An Emerging Frontier for Diabetes and Diabetic Complications Therapy”. The article is well structured and the general idea appears attractive. Furthermore, the use of MSC-derived extracellular vesicles for the treatment of diabetes represents a novelty in the field. However, there are many points that should be addressed before the publication of this article:

MAJOR POINTS:

Q: Authors should clarify which is the clinical benefit of MSCs-EVs therapy for Diabetes Mellitus and their clinical complications in comparison to current therapies such as therapeutic insulin and hypoglycemic agents. They could specify the percentages of clinical improvement to assess the real benefit of this treatment.

Response: Thanks for your suggestion. We have added a section to clarify the clinical benefit of MSC-EVs therapy for DM and diabetic complications in comparison to therapeutic insulin in Line 366-384 of the revised manuscript. Current studies on MSC-EVs-mediated therapy for DM and diabetic complications are still in preclinical stage. There are few studies to compare MSC-EVs with clinical therapeutic strategies. We have listed two researches which reveal that MSC-EVs exhibit more effective roles to alleviate DM and diabetic nephropathy compared with insulin treatment. However, the comparison between MSC-EVs and current therapies in other diabetic complications has not been reported.

Q:  For the third section it could be interesting to organize the different preclinical studies to improve understanding of the text. Apparently, it makes more sense to explain the preclinical studies based on similar results obtained in order to appreciate the major benefits.

Response: Thanks for your suggestion. We have re-organized the preclinical studies based on similar results in the third section of the revised manuscript.

In DM paragraph, we have illustrated the therapeutic effects of MSC-EVs through the mechanisms including immune regulation, β-cell protection and peripheral insulin resistance remission.

In diabetic nephropathy paragraph, we have illustrated the renal therapeutic effects of MSC-EVs in STZ-induced diabetic animal models and db/db mice firstly, and then clarified their repairing roles in specific renal cells such as podocytes and epithelial cells.

In diabetic retinopathy paragraph, we have illustrated the retinal therapeutic effects of MSC-EVs in vitro and in vivo firstly, and then clarified the potential mechanisms and functional components of MSC-EVs in the treatment of diabetic retinopathy.

In diabetic wound healing paragraph, we have illustrated the therapeutic roles of MSC-EVs to promote wound closure, re-epithelialization, collagen deposition, and neovascularization firstly, and then clarified the potential mechanisms and functional components of MSC-EVs in diabetic wound healing.

In diabetic neuropathy paragraph, we have illustrated the therapeutic roles of MSC-EVs in neuroprotection and cognitive function recovery respectively.

MINOR POINTS:

Q: In lines, 52-54, as well as in the section 2, authors mention the classification of EVs, however, I would recommend authors to follow the 2018 MISEV guidelines of the ISEV and provide a more rigorous classification. Authors should start from Small and Large EVs and based on the markers and pathways followed continue with an in depth classification.

Response: Thanks for your suggestion. We have provided a more rigorous classification of EVs according to the 2018 MISEV guidelines of the ISEV in Line 75-82 of the revised manuscript. “Since the specific markers of EV subtypes remain further discussion and investigation, the International Society for Extracellular Vesicles (ISEV) recommends researchers to use operational terms for defining EVs according to physical characteristics (size and density), biochemical composition, or descriptions of conditions or cell of origin [24]. For instance, EVs can be divided into small EVs (<100 nm or <200 nm) and medium/large EVs (>200 nm). Previous studies have usually classified EVs into exosomes, microvesicles, and apoptotic bodies [25]. However, strictly speaking, they should only be used when data revealing the formation processes of EVs is provided.”

Q: In lines 96-98 we recommend mentioning the role of lipids in the biogenesis process as it appears in figure 1 for a better comprehension of the text.

Response: Thanks for your suggestion. We have provided the role of lipids in the biogenesis process in Line 107-109 of the revised manuscript. “Lipid-raft domains of the plasma membrane contribute to the early endosome formation through the endocytosis pathway, thus initiating the biogenesis process of exosomes [36].”

Q: The explanation of figure 1 could appear between lines 128-129 in order to enhance a correct follow-up of the text.

Response: Thanks for your suggestion. We have moved the figure 1 and its explanation to the correct position.

Q: In lines 72 and 182 we have detected two orthographic mistakes. Before and is not necessary to include a coma and the name of Sun et al., should be in capital letters.

Response: Thanks for your reminder. We have corrected these orthographic mistakes in Line 73 and 190 of the revised manuscript.

Q: In line 203 appears the biological molecule GW4869 please include an explanation for better comprehension.

Response: Thanks for your suggestion. We have provided an explanation about GW4869 in Line 230-231 of the revised manuscript. “GW4869 (an inhibitor of EVs secretion)”

Q: The conclusion could be extended by explaining the clinical breakthrough that MSCs-EV could bring in the Diabetes Mellitus field.

Response: Thanks for your suggestion. We have extended the conclusion by explaining the enhanced therapeutic effects of MSC-EVs in the treatment of DM and diabetic complications by comparing with current therapies in Line 590-600 of the revised manuscript. “Current clinical treatment for DM mainly relies on the injection of insulin and some hypoglycemic agents. However, this strategy can only temporarily control blood glucose and even cause side effects such as subcutaneous nodule, diarrhea, and obesity [174]. Moreover, exogenous insulin administration may aggravate β-cell dysfunction, leading to the development of peripheral insulin resistance and diabetic complications [175]. Various preclinical studies have demonstrated that MSC-EVs show effective roles to alleviate hyperglycemia in diabetic animal models by improving β-cell mass, promoting insulin sensitivity, and increasing glucose uptake and metabolism in peripheral tissues. In addition, MSC-EVs also represent promising candidates to ameliorate diabetic complications through multiple mechanisms. The application of MSC-EVs brings a key breakthrough in the treatment of DM and diabetic complications.”

We hope that the above responses meet the expectations of the reviewer. The comments and suggestions are helpful in improving the quality of our manuscript.

Thank you for your consideration. We look forward to hearing from you and to publication of this manuscript.

Sincerely,

Hui Qian, Prof.

School of Medicine, Jiangsu University,

301 Xuefu Road, 212013, Zhenjiang, Jiangsu, P.R. China.

E-mail: lstmmmlst@163.com

Reviewer 4 Report

Sun et al. reviewed the current research on the potential therapeutic use of MSC-derived EVs for diabetes and diabetic complication therapy. The content is timely and significant in the EV and diabetes research field. However, the manuscript needs more clarification and accurate information. The detailed comments are listed below. 

  1. Line 86, please add a more accurate description to “membrane transport and fusion protein.” Also, please specify which RAB protein, GTPases, and annexins. 
  2. Line 102, cellular uptake of EVs involves various endocytosis pathways, such as receptor-dependent (clathrin-mediated, caveolin-mediated) or -independent endocytosis (macropinocytosis). Please refer to the review, Mulcahy et al., JEV, 2014, doi.org/10.3402/jev.v3.24641. 
  3. Line 209, please add a brief explanation of podocytes for non-expert readers. 
  4. Line 228, please add a more detailed description to improve the clarity. Does miR-26a-5p decrease the expression of TLR4 to suppress immune activation? 
  5. Line 253, please add a more detailed description to improve the clarity. Does NEDD4 ubiquitinate PTEN for degradation? How is this related to the downstream AKT/NRF2 pathway?
  6. Line 262, the authors stated “the survival time of MSC-EVs in the vitreous”. Please add a more accurate description. How were the MSC-EVs cleared, by cellular uptake or excluded from the body? If it is taken up by cells, it is not simply cleared, but there may be some effects on the recipient cells. 
  7. LIne 278, please clarify the “tube formation abilities of diabetic wound.”
  8. Line 348, please clarify the possible mechanism of the HOTAIR in mediating angiogenic effects. 
  9. Line 363, please add brief explanations of resveratrol, pioglitazone, atorvastatin, deferoxamine, and melatonin for non-expert readers.
  10. Line 406, this paragraph listed studies that used various hydrogels. The information is important but it is hard to understand the individual studies. Please consider including a table listing these studies as table 1. 
  11. Line 481, the authors stated that developing a new separation approach is important. However, there is no section describing the current isolation methods of MSC-EVs. Please consider including a section on the advantages and limitations of the current MSC-EV isolation methods or citing other reviews on this.
  12. Line 483, similarly, the significance of 3D culture systems is stated without describing in detail on this. Please consider including a section on this or citing other reviews on this. 

Author Response

Oct 10, 2022

Pharmaceutics

Manuscript ID: pharmaceutics-1951709

Title: Mesenchymal Stem Cell-Derived Extracellular Vesicles: A Potential Therapy for Diabetes and Diabetic Complications

Authors: Fengtian Sun, Yuntong Sun, Feng Wu, Wenrong Xu *, Hui Qian *

Dear Editor,

Thank you for your e-mail dated Oct 2, 2022 regarding the review of our manuscript. We appreciate your assessments of the manuscript and have found that the comments and suggestions are helpful in preparation of the revised manuscript. We revised the manuscript as suggested by reviewers. In the revised manuscript, all the amendments were highlighted by yellow words. The following are our point-by-point responses, in order of the comments about the manuscript:

Sun et al. reviewed the current research on the potential therapeutic use of MSC-derived EVs for diabetes and diabetic complication therapy. The content is timely and significant in the EV and diabetes research field. However, the manuscript needs more clarification and accurate information. The detailed comments are listed below

Q: Line 86, please add a more accurate description to “membrane transport and fusion protein.” Also, please specify which RAB protein, GTPases, and annexins.

Response: Thanks for your suggestion. We have added a more accurate description and specified RAB protein, GTPases, and annexins in Line 95, 96 of the revised manuscript. “proteins involved in membrane fusion (Rab5, Rab7, Annexin A1, Annexin A2, and Annexin A7)”

Q: Line 102, cellular uptake of EVs involves various endocytosis pathways, such as receptor-dependent (clathrin-mediated, caveolin-mediated) or -independent endocytosis (macropinocytosis). Please refer to the review, Mulcahy et al., JEV, 2014, doi.org/10.3402/jev.v3.24641.

Response: Thanks for your suggestion. We have read this review and provided a more detailed description about cell uptake pathway of EVs in Line 113-117 of the revised manuscript. “Among them, various endocytosis-related ways are considered as the major mechanisms underlying the uptake of EVs by recipient cells, including clathrin-dependent endocytosis, caveolin-mediated uptake, macropinocytosis, phagocytosis, and lipid raft-mediated internalization [39].”

Q: Line 209, please add a brief explanation of podocytes for non-expert readers.

Response: Thanks for your suggestion. We have added a brief explanation of podocytes in Line 244-247 of the revised manuscript. “As one type of terminally differentiated epithelial cells, podocytes exert critical roles in maintaining glomerular filtration barrier function [83]. Hyperglycemia-caused podocytes injury is recognized as the key pathogenic factor of glomerular hyperfiltration and proteinuria during diabetic nephropathy progress [84].”

Q: Line 228, please add a more detailed description to improve the clarity. Does miR-26a-5p decrease the expression of TLR4 to suppress immune activation?

Response: Thanks for your suggestion. We have added a more detailed description in Line 252-254 of the revised manuscript. “Duan et al. have revealed the protective role of miR-26a-5p from MSC-EVs to enhance the viability of podocytes by reducing Toll-like receptor 4 (TLR4) expression and inhibiting downstream NF-κB/VEGF pathway [87].”

Q: Line 253, please add a more detailed description to improve the clarity. Does NEDD4 ubiquitinate PTEN for degradation? How is this related to the downstream AKT/NRF2 pathway?

Response: Thanks for your suggestion. We have added a more detailed description in Line 281-286 of the revised manuscript. “For example, the results of Sun et al. have shown that hyperglycemia-induced upregulation of phosphatase and tensin homolog (PTEN) can inhibit AKT phosphorylation and nuclear factor erythroid 2-related factor 2 (NRF2) expression in retinal tissues, while MSC-EVs ameliorate retinal apoptosis and oxidative stress by transporting neuronal precursor cell-expressed developmentally downregulated 4 (NEDD4) to cause PTEN ubiquitination and degradation and activate downstream AKT/ NRF2 pathway [95].”

Q: Line 262, the authors stated “the survival time of MSC-EVs in the vitreous”. Please add a more accurate description. How were the MSC-EVs cleared, by cellular uptake or excluded from the body? If it is taken up by cells, it is not simply cleared, but there may be some effects on the recipient cells.

Response: Thanks for your suggestion. We have added a more accurate description in Line 292-297 of the revised manuscript. “However, recent evidence has revealed that intravitreally administered MSC-EVs are rapidly absorbed by superficially located cells such as astrocytes, microglia, and retinal neurons, leaving few MSC-EVs to penetrate deeper into the retina [103]. Notably, the retention time of MSC-EVs in retinal cells is less than 14 days. It is necessary to develop novel strategies to maintain MSC-EVs in retinal tissues for achieving increased cell uptake and extended residence time.”

Q: LIne 278, please clarify the “tube formation abilities of diabetic wound.”

Response: Thanks for your suggestion. We have provided a more accurate description in Line 309-312 of the revised manuscript. “Moreover, Liu et al. have revealed that MSC-EVs can enhance the proliferation, migration, and angiogenesis abilities of vascular endothelial cells by promoting hypoxia-inducible factor-1α (HIF-1α) expression in PI3K-AKT-mTOR dependent manner, leading to the acceleration of diabetic wound healing [111]”

Q: Line 348, please clarify the possible mechanism of the HOTAIR in mediating angiogenic effects.

Response: Thanks for your suggestion. We have clarified the possible mechanism of the HOTAIR in mediating angiogenic effects in Line 405-407 of the revised manuscript. “HOX transcript antisense RNA (HOTAIR) has been reported to play critical roles in mediating angiogenic effects of endothelial cells by enhancing VEGF expression and reversing miR-761-induced histone deacetylase 1 inhibition [129, 130].”

Q: Line 363, please add brief explanations of resveratrol, pioglitazone, atorvastatin, deferoxamine, and melatonin for non-expert readers.

Response: Thanks for your suggestion. We have added brief explanations of resveratrol, pioglitazone, atorvastatin, deferoxamine, and melatonin in Line 421-437 of the revised manuscript.

Q: Line 406, this paragraph listed studies that used various hydrogels. The information is important but it is hard to understand the individual studies. Please consider including a table listing these studies as table 1.

Response: Thanks for your suggestion. We have added a table named Table 3 to further illustrate the studies involving hydrogels in Line 520 of the revised manuscript.

Q: Line 481, the authors stated that developing a new separation approach is important. However, there is no section describing the current isolation methods of MSC-EVs. Please consider including a section on the advantages and limitations of the current MSC-EV isolation methods or citing other reviews on this.

Response: Thanks for your suggestion. We have added a section and a table to clarify the advantages and limitations of the current MSC-EVs isolation methods in Line 144-153 of the revised manuscript.

Q: Line 483, similarly, the significance of 3D culture systems is stated without describing in detail on this. Please consider including a section on this or citing other reviews on this.

Response: Thanks for your suggestion. We have added a section to descript the 3D culture systems in Line 462-480 of the revised manuscript.

We hope that the above responses meet the expectations of the reviewer. The comments and suggestions are helpful in improving the quality of our manuscript.

Thank you for your consideration. We look forward to hearing from you and to publication of this manuscript.

Sincerely,

Hui Qian, Prof.

School of Medicine, Jiangsu University,

301 Xuefu Road, 212013, Zhenjiang, Jiangsu, P.R. China.

E-mail: lstmmmlst@163.com

Round 2

Reviewer 1 Report

The authors addressed all of the comments and revised the content of the manuscript accordingly. In light of this, I recommend that it may be considered for publication in Pharmaceutics Journal in its present form.

Reviewer 4 Report

The authors addressed all of my comments in the response letter and significantly improved the manuscript. I suggest the authors consider a professional English editing service to improve the readability. Other than that, I recommend the revised manuscript for publication.